# Adherence to the Mediterranean Diet Related to the Health Related and Well-Being Outcomes of European Mature Adults and Elderly, with an Additional Reference to Croatia

**DOI:** 10.3390/ijerph20064893

**Published:** 2023-03-10

**Authors:** Manuela Maltarić, Paula Ruščić, Mirela Kolak, Darija Vranešić Bender, Branko Kolarić, Tanja Ćorić, Peter Sousa Hoejskov, Jasna Bošnir, Jasenka Gajdoš Kljusurić

**Affiliations:** 1Andrija Štampar Teaching Institute of Public Health, Mirogojska 16, 10000 Zagreb, Croatia; 2Faculty of Food Technology and Biotechnology, University of Zagreb, Pierottijeva 6, 10000 Zagreb, Croatia; 3School of Medicine, University of Zagreb, Šalata 2, 10000 Zagreb, Croatia; 4Unit of Clinical Nutrition, Department of Internal Medicine, University Hospital Centre Zagreb, Kišpatićeva 12, 10000 Zagreb, Croatia; 5WHO Health Emergencies Programme, World Health Organization Regional Office for Europe, UN City, Marmorvej 51, DK-2100 Copenhagen, Denmark

**Keywords:** health-related behaviors (HBR), food intake, well-being, elderly, Mediterranean diet

## Abstract

With the increase in life expectancy, expectation of a longer healthy life is also increasing. Importance of consumption of certain foods is confirmed to have a strong effect on quality of life. One of the healthiest dietary patterns consistently associated with a range of beneficial health outcomes is the Mediterranean diet (MD). The aim of this study was to assess MD adherence in the population over 50 years of age, in Europe, with special reference to Croatia, and to assess regional differences and investigate the association with health-related indicators (disease incidence, body mass index (BMI), grip strength measure, control, autonomy, self-realization, and pleasure scale (CASP-12)). This research uses data from the SHARE project for the population over 50 years of age. The frequency of individual responses was analyzed (frequencies, cross tables, and appropriate tests of significance, depending on the data set), and logistic regression was used to connect adherence to the Mediterranean diet with health indicators. The results of the study indicate a positive correlation between adherence to the principles of the Mediterranean diet with CASP and self-perception of health, which the followers of the MD pattern predominantly rate as “very good” or “excellent” (37.05%) what is significantly different (*p* < 0.05) from individuals which do not follow the patterns of MD (21.55%). The regression models indicate significant changes in the measure of maximum grip strength also among MD followers (OR_MEDIUM_ = 1.449; OR_HIGH_ = 1.293). Data for EU countries are also classified by regions (Central and Eastern; Northern, Southern and Western Europe), additionally allocating Croatia, and the trends in meat, fish and egg consumption showed the greatest differences for Croatian participants (39.6% twice a week) versus participants from four European regions. Data for Croatia deviates from the European average also in terms of the proportion of overweight and obese persons in all observed age groups, of which the largest proportion is in the 50–64 age group (normal BMI: only 30.3%). This study extended the currently available literature covering 27 European countries, placing the findings in a wider geographical context. The Mediterranean diet has once again proven to be an important factor related to health-related behavior. The presented results are extremely important for public health services, indicating possible critical factors in preserving the health of the population over 50 years old.

## 1. Introduction

One of the critical public health issues in the elderly population is considered to be health-related behavior (HRB) in which diet is a central health-related behavior that needs to be investigated along with factors such as physical activity [1]. With the increase in the development of European countries, life expectancy is also increasing, as well as the expectation of a longer healthy life [2] for men 82.9 years and for women 85.1 years. The share of people over 65 years of age in the EU population was 20.6% (in 2020), and it is expected that the share of this population group will increase to 40.6% by 2050 [3]. Recently conducted research [1,4] indicates an increase in the number of sick people in general, and especially in the group of the elderly population, which among the many causes singles out the reduced healthy behavior of individuals, and thus the weaker general well-being of the individual.

In 2019, there were 42.28% of people over 50 years old in the Republic of Croatia [5]. In 2021, the aging of the population continued, and the average age of the total population of the Republic of Croatia was 44.3 years (men 42.5 years, women 45.9 years), which ranks it among the oldest nations in Europe [6]. The highest share of the population aged 65 and over, in relation to the total population, was in Šibenik-Knin County (27.3%) and in Lika-Senj County (26.2%), and the lowest in the city of Zagreb (20.6%) and in Međimurje County (20.5%). According to Eurostat’s Nomenclature of territorial units for statistics (NUTS), the mentioned counties are two different regions, so the first two counties are classified as Adriatic Croatia, and the last two as continental Croatia [7]. NUTS-2 regions between 2012 and 2020 were Adriatic and Continental Croatia, and since 2021, the Continental Croatia was additionally divided to three regions: Pannonian Croatia, Adriatic Croatia and Northern Croatia [8]. At the age over 50, and especially over 65, the rate of getting sick from various diseases increases [9]. Based on the Eurostat data, among the most common diseases are ischemic heart disease, malignant diseases of the respiratory system, diabetes, cirrhosis of the liver, cerebrovascular insult, malignant cancers of the colon and rectum, COPD, dementia, various kidney diseases and hypertensive heart disease. As chronological age increases, various changes occur at the molecular level that affect the entire organism in various ways [3]. However, in all population groups, including the population over 50 years of age, the proportion of overweight and obese people has increased. Due to increased food intake, hypertriglyceridemia and hypercholesterolemia occur, which favors the development of atherosclerosis of blood vessels and increased frequency of myocardial infarction and as a result of compensatory hyperinsulinemia, which occurs due to a decrease in the sensitivity of insulin receptors, diabetes occurs [10]. It is also known that in certain age increases the share of malnourished population. Malnutrition most often occurs either as a primary disease due to increased loss of nutrients or decreased consumption of foods or as a result of other diseases where is of particular concern the cachexia–state of extreme malnutrition and physical and metabolic exhaustion due to a serious illness [11]. According to all previously mentioned, it is clear that the increased involvement of public health services is due to the extension of life expectancy, setting as the main goal of the services to increase the number of years of healthy life and improve the quality of life in those years. In addition to public health service, prevention strategies may also be needed including the establishment of an enabling physical and institutional environment for a healthy living.

Importance of consumption of certain food groups is confirmed to have a strong effect on quality of life [4,12] where consumption of fruits and vegetables is singled out to be positively associated with subjective health and quality of life, along with physical activity and good sleep where the middle-age adults is the better sleep more relevant while in the older population it is the consumption of fruits and vegetables [12]. Among the dietary patterns in which the emphasis is on the intake of fruits and vegetables, the Mediterranean diet (MD) stands out [13] and features of this diet are everyday consumption of fruits, vegetables, olive oil, wholegrain breads, cereals, herbs, spices and frequent intake of eggs, beans or legumes, fish and seafood. Studies show that MD adherence is associated with a lower risk of a number of infectious diseases and better health outcomes [14,15], which is particularly important for the population group of mature and elderly people. However, in the population over 50, there is an increased risk of impaired health, which increases proportionally with age. Vall Castello and Tubianosa [13] used the method of Mediterranean diet adherence (MD) in Europe, concluding on the population of adults aged 50 years and above that following the Mediterranean diet was negatively correlated with the incidence of chronic illnesses, as well as with levels of depressive symptoms, conducted on data collected during 2011–2017. The main objectives of this study are to investigate the Mediterranean adherence and its relation with the health-related indicators (disease incidence, maximum of grip strength measure, CASP-12 (control, autonomy, self-realization, and pleasure scale) and body mass index)) in the middle-aged and elderly population in Europe with a special emphasis on participants from Croatia (from the SHARE base). Results for Croatia, from the database SHARE are being presented for the first time. The objectives are focused to test and verify: (i) if relationships exist between adhering to the Mediterranean diet, or not (food intake and the frequency of consumption of certain food groups) and variations in health outcomes of individuals (as the incidence of chronic diseases), their levels of body mass index (BMI) and the levels of self-assessed well-being; (ii) if correlations can also be observed between physically active and healthy lifestyle with the same health outcomes.

All objectives will be tested on a large database that includes 27 European countries and if the mentioned connections prove to be significant, they will have exceptional value for all participants in food production and processing chain up to the food on the table, intended for the population included in this study. For the first time, data for the Republic of Croatia were additionally separated for this population group.

## 2. Materials and Methods

### 2.1. Study Design

This study uses data from the SHARE project (Survey of Health, Ageing and Retirement in Europe), wave 8 (2019), release 8.0.0 [16]. SHARE is a European multidisciplinary and cross-national panel database of microdata on health, socioeconomic status and social- and family networks. Croatia participates in SHARE since 2004 [17], and SHARE data set contains representative samples of the 50+ population in each European country, plus Israel. Data is collected in face-to-face interviews using the computer-assisted personal interviewing (CAPI) method. Proxy interviews are allowed when respondents are unable to participate in the survey, such as health reasons. For further methodological details of the SHARE project, please see [18]. In wave 8 participated 27 European countries (Austria, Germany, Sweden, Netherlands, Spain, Italy, France, Denmark, Greece, Switzerland, Belgium, Israel, Czech Republic, Poland, Luxembourg, Hungary, Slovenia, Estonia, Croatia, Lithuania, Bulgaria, Cyprus, Finland, Latvia, Malta, Romania and Slovakia) and Israel participated in the SHARE project. From the total sample size (*n* = 114,199) were excluded data for participants under 50 years (0.5%, *n* = 236). The baseline indicators of the included population group are presented in Table 1.

### 2.2. Mediterranean Diet

This survey contains questions that are intended to assess the quality of the dietary patterns of the participants. The quality of the diet was assessed using these questions:How often do you eat a serving of dairy products which means a cup of yogurt, cheese, a glass of milk, or a can of high protein supplements weekly?How often do you eat a serving of eggs, beans, or legumes weekly?How often do you eat a serving of poultry, meat, or fish weekly?How often do you eat a serving of fruits and vegetables weekly?

The participants could choose following answers:Less than once a week;Once a week;Twice a week;3–6 times a week andEvery day.

In the study of Alves and Perelman [14], it was indicated that with this data, it is possible to assess whether the participants followed the principles of the Mediterranean diet which is recognized as a recommended diet pattern. Although the Mediterranean diet is accompanied by the consumption of whole grain products, olive oil/olives, moderate wine consumption, according to previously published works that used data from the SHARE database, a recommendation was made based on 4 researched food groups [13,14,15]. Mediterranean diet is based on the everyday consumption of veggies and fruits and frequent intake of eggs, beans, legumes, meat, fish, or poultry (three-six times weekly [13,14]). From the collected data of the frequency of the before mentioned 3 food groups (not including the dairy) we have created a binary index identifying which of the participants may be classified as subscribing to a Mediterranean diet (value = 1 if following the diet; 0 if not following the diet). Limitation of this database is that it does not separate red meat from the other types of meat [15] but results of this study will be compared to those studies that had pointed out the same limitation. A summary of the frequency of food consumption from four food groups is shown in Table A1 (note—the consumption of dairy products is not included in the calculation of MD indicators).

The main limitation of the above-mentioned estimation of MD adherence is the partial analysis due to the imprecise characterization of this dietary pattern. By such assessing of MD adherence, some key foods such as seeds and oilseeds, the consumption of olive oil, red wine and seafood [17], and the separation of the type of meat (red meat, cured meat products) are not included.

### 2.3. Health Related Indicator

Two parameters were used as an indicator of population health, the first of which is related to nutrition–the body mass index, and the second is related to the most common cardiovascular and metabolic diseases (CMDs). The disease indicator was generated according to the study by [13] and a chronic disease variable was generated that included diseases related to cardiovascular diseases (myocardial infarction and other heart problems, hypertension, high blood cholesterol, stroke) and type 2 diabetes. Possible outcomes for this variable are 0 (absence of any of the mentioned diseases) and a maximum of 5, which indicates that the respondent stated that he suffers from all the mentioned diseases. Incidence of Cardio Metabolic Diseases for the population included in this study is shown in Table A2. The body mass index was calculated from the data on the body weight and body height of the subjects, and the final values were grouped into the groups malnourished, normally nourished and overweight and obese depending on age. For the age group up to 65 years, values greater than 25 indicate excess body mass and obesity [19], while according to the ESPEN guidelines for people older than 65, the limit has been moved and the range of normal nutrition is 21–27.5 kg/m^2^ [20]. The classification itself and the basic overview of the population group is given in Table A3 and Table A4. 

### 2.4. Maximum of Grip Strength Measure

Hand grip strength (HGS) is the maximum static force applied by the hand and it is measured by a dynamometer. It is simple and inexpensive to evaluate. Furthermore, this measurement is widely applied not only to assess hand function after injury and the outcome of hand surgery, but also to reflect general physical health and disability [21,22,23]. Lower HGS as concluded in a longitudinal study was connected to increased mortality risk from cardiovascular diseases and cancer, even after adjusting for body composition, multiple chronic diseases, and multi-morbidity, in both sexes [21,24].

### 2.5. CASP-12

Two parameters were used as an indicator of population health, the first of which is

CASP-12 (control, autonomy, self-realization, and pleasure scale) is one of the most common internationally used measures for quality of life in older adults, although its structure is not clearly established [25]. This scale is theoretically driven by the ‘needs satisfaction’ approach to measure QoL in early old age. It is based on Maslow’s Hierarchy of Needs [26]. The CASP-12 scale is a modification of the original CASP-19 [27]. Authors from study [28] conducted analyses on all countries in SHARE Wave 4 and concluded that the theoretical four-factor structure of the CASP did not meet the data properly. Another author [29], employing Item Response Theory (IRT) models, found a bifactor model with a strong global factor of QoL to better represent CASP-12 scores, with data used from SHARE Wave 6 [25]. Nevertheless, other studies have found good fit for the four-factor theoretical structure in the CASP-12 [25].

### 2.6. Statistical Analyses

Before the actual data processing, the missing data analysis was performed. For this cross-sectional study, the first step was to analyze missing data by country, because in the study by [30]. pointed out that data analysis will be credible if the proportion of missing values is less than 5%, which is not the case for some variables (e.g., economic and health indicators). Therefore, the procedure of a study [31] applied in this paper as well, and it suggests multiple imputation of missing values to increase the number of observations. After including these imputed variables, missing data remained (less than 1%, per country). 

In the SHARE documents are available coded data with nominal and scale variables. Based on the data type, different tests were applied to assess the differences in the observed characteristics of the examined population, statistical tests were performed to compare the two groups (*t*-test (t) and chi-square test (χ^2^)). Individuals following the Mediterranean diet (MD) were compared with those individuals who did not follow it.

Data analysis was performed with covariates which included; age (ranged from 51–104), gender (female/male), marital status (all married or living with the partner were categorized as “living with partner” and the widowed, divorced, etc. are classified in the group “Not living with partner”), education level (primary (9 years of education); secondary (maximal 12 years of education) and tertiary (more than 12 years of education), economic status (household able to meet ends meet (with difficulty is defined as poor; fairly easily as fair and easily as good), employment (retired = retired while other categories were defined as not retired), self-perceived health (poor and fair= Poor/Fair, good = Good and very good and excellent is grouped as Very good/Excellent), the CASP index for quality of life and well-being (12–24 = Low; 25–36 = Medium; >36 = High), Sport or activities that are vigorous (Never or rare, At least once a week summarized all activities including at least activity once a week), CMD (0 = none, 1–5 = one or more diagnoses), BMI (those who fall into the category of normally nourished = normal, depending on the age group (18.5–25 kg/m^2^ for the aged under 65, and 21–27.5 for those over 65 years) and the max. of grip strength measure (1–25 = Low; 26–50 = Medium and >50 = High).

Logistic regression was applied to explore variable changes related to adherence of Mediterranean diet (following vs not following). Crude regression model (CRM) observed difference based on the MD, while the bivariate-adjusted model included age, gender, and marital status (M1). To remove their influence on the adherence of MD, the next bivariate-adjusted model (M2) included Model 1 observations and education level, economic status, and employment. Odds ratios (ORs) with corresponding 95% confidence intervals (CIs) were estimated for all models (Crude, Model 1, and Model 2). OR > 1 indicates a greater likelihood of an association with the exposure and outcome or an increased incidence of the investigated event. The 95% confidence interval for the OR is given in square brackets, and statistical significance was tested with the chi-square test. Data analysis software SPSS v. 19 was used.

## 3. Results

Before the conducted regression analysis, the baseline characteristics of the population group were determined and presented in Table 1 where the results are presented as frequencies (%), for each observed variable.

On Figure 1A can be seen that the mean values of maximum grip strength for the participants who followed MD mostly show lower values than the participants who were not following MD, with a few peaks in the other direction. Contrary to this phenomenon, on Figure 1B is present an almost inverted situation meaning that the participants who were not following MD show lower mean CASP index for quality life and well-being than the participants who were following MD.

In the study of [13] is already confirmed that the incidence of CMDs is slightly lower in those who follow the MD, so in Figure 1 are presented the similarities/differences in the max. of grip strength measure and index for quality of life and well-being for those participants who follow (or not follow) the patterns of the Mediterranean diet. Those results clearly indicate that participants following MD in general have higher CASP.

There are few observations that can be spotted from Table 1. First, we can see that the distribution of “Not following MD” and “Following MD” among age and sex of the participants is almost equal, as well as the marital status, educational level and employment. The only variable that stands out from these “input” data is economic status, which is greater for the participants who were not following MD. This is quite self-explanatory because the majority of people cannot afford MD since it more expensive This statement is a result of a study published by Rubini and coworkers [32] which included 2833 subjects, concluding that the degree of adherence to the MD eating pattern was positively correlated with the monthly cost, emphasizing that the required economic effort to afford the MD is higher [32]. Self-perceived health is the highest for the participants following MD and lowest for participants not following MD which means that the participants are aware of the benefits the MD offers. Sports activities and CMD consequently followed by body mass index also presents higher percentage at the “Following MD” participants, while maximum of grip strength measure show slightly bigger values among “Not following MD” participants. This also matches the data observed from Figure 1.

Croatian participants included in the study present 2.5% valid samples used for the analyses (*n* = 1197). The characteristics based on observed significant variables for the Croatians are presented in Table 2.

The data obtained from Table 2 show almost equal input parameters as Table 1, but with the segregation of the participants among the Adriatic and Continental parts of the country. Additionally, the characteristics of the population as age, gender and marital status, are not equally distributed among groups, so the conclusion drawn out from this table can only by partially taken. At this age, the mortality of the male population is higher [28] and the majority of participants who stated that they did not live with a partner were widowed [14,25]. The most notable observations can be spotted in the CASP parameter: no participant from Adriatic part following MD declared low values, while the highest values were observed from both Adriatic areas (“Following” and “Not following MD”). Maximum of grip strength measure was noted 0 at the Following MD Adriatic participants. 

The data from Table 3 tell us that the most participants in these conducted experiments fit in the overweight BMI category with almost 40%, while most of them aging between 65 and 74 years. The lowest frequency with less than 1% can be seen at the underweight category. 

Means value of the BMI was 26.25 kg/m^2^ with a standard deviation of ±6.85 kg/m^2^ and in Table 4 are presented the shares of the population with the normal BMI according to the corresponding age group, per countries.

The most notable observation from Table 4 is that the countries with the highest BMI from 50 to 64 years of age are Switzerland, France and Netherlands with values 48.4, 42.2 and 51.8, respectively. BMI from 65 to 74 years of age are participants from Sweden, Italy and Switzerland with 61.2, 59.7 and 57.3, respectively. BMI for the participants aging between 75 and 84 years are obtained for Sweden, Denmark and again Switzerland, with 62.4, 59.8 and 58.5, respectively. Lastly, the highest BMI values for the participants older than 85 years are obtained for Netherlands, Germany and Austria, with values 63.7, 61.8 and 60.7, respectively. The lowest values of BMI for all age groups are spotted for Malta with 14.6, 23.8, 30.2 and 24.4, respectively.

Figure 2 shows trends in nutrition of the European population by age and additionally by gender in the Republic of Croatia. There is a clear trend of a higher proportion of overweight in the male population, regardless of age group, and the proportion of obese people is more pronounced in the female population. An increase in the proportion of malnourished people with increasing age is also evident. The Croatian population, which is in an unenviable, leading position in terms of the proportion of overweight and obese in the population, was additionally analyzed with the aim of determining the age groups in which additional education/intervention is necessary (Figure 2B).

Finally, the logistic regressions were used to investigate the Mediterranean diet adherence (Following MD) in relation of variables that are associated to the health and well-being of the investigated participants (self-perceived health (SPH), index for quality of life and well-being (CASP), vigorous sport activities, CMDs, body mass index (BMI) and the maximum of grip strength measures). The models were adjusted for age, gender, marital status, education level, employment and economic status and the final result is presented in Table 5 Adjusted odd ratios indicated that with a lower SPH assessment, the adherence to the principles of the Mediterranean diet is lower (OR = 0.921), in contrast to those who rated it as good (OR = 0.927) and especially those with a rating of very good and excellent (OR = 1.541). The CASP index for quality of life and well-being showed a significant increase in following the MD (OR_Low_ = 1.095 vs. OR_High_ = 1.195, *p* < 0.01). For respondents who regularly engage in vigorous sport activities the relation with the MD adherence is significant (OR_regular_ = 0.910 vs. OR_Never_ = 0.942, *p* < 0.01). The cardiovascular and metabolic diseases variable indicated higher MD adherence if the participants have at least one of the diagnoses (OR_none_ = 0.981 vs. OR_Never_ = 1.019). Regardless of which age group it is, the proportion of those with a normal BMI (up to 65 divisions according to standard tables, after 65 according to ESPEN guidelines) was singled out, the connection with following the guidelines of the Mediterranean diet is evident, indicating that in most of those who following the principles of the Mediterranean diet, a normal BMI is expected (OR = 1.054 with a CI = 0.993 to 1.119). The maximum of grip strength measure showed significant increase for low to high outcomes (OR_Low_ = 0.774 vs. OR_High_ = 1.293). Table A4 shows the frequency of food consumption from specific groups for four European regions as well as for two Croatian regions, while Table A5 shows the systematized incidence of diseases in the population over 50 years of age, also presented according regions [7,8,32]. Southern Europe countries dominate in the every day intake of dairy products, legumes, beans, eggs and meat, fish and poultry, while for the last food group, fruits and vegetable consumption on the everyday basis, the first place is shared with the countries of Western Europe (74.5%). Statistically significant differences were indicated in the frequency of consumption of fruits and vegetables in the Northern European countries (61.7% *p* < 0.05) what is in accordance with the different nutritional habits in EU countries [33]. Incidence of observed diseases which were available in the SHARE database were systematically presented according four European regions with use of heatmap principles and their indication of high or low values by use of colors. Southern European countries have the lowest proportions of patients with 50% of all twenty observed health issues (marked in dark green). Western countries have the highest proportion of population for whom depression was ever diagnosed/currently having (43% vs. 33.6 % in Southern European countries), as well as the Osteoarthritis/other rheumatism where the incidence is significantly higher (25.9% vs. 19.4% in Central and Eastern countries; 18.8% in Northern countries and 15.2% in Southern European countries, respectively). Croatia follows the trend of Central and Eastern Europe related to the number of people suffering from the mentioned diseases (Table A5). 

## 4. Discussion

Presented results confirmed the relationship of the adherence MD and the health and well-being outcomes. 

The results firstly indicate the parameter as maximum of grip strength measure did not indicate significant relation with the adherence MD (Figure 1A) while the logistic regression models indicatted there was a significant relationship (Table 5). This is an outcome confirmed by Tak and coworkers [21] which investigated the relation of the grip outcome and diet. Different studies investigated the relationship between the CASP and adherence of MD [9,13]. Our findings confirm the proportional relation of the CASP outcomes to MD diet adherence, so if a person has a higher CASP, it is more likely that they will also follow the principles of the Mediterranean diet. 

Characteristics of the population related to following the MD indicated statistically significant differences based on gender (in favor of female population), living with the partner, having fair economic status and high CASP index, as well as for those participants who self-perceived their health as very good/excellent. Similar findings were confirmed with the difference that CMDs were in negative relation to MD followers [13].

Croatia specific geographic location (HR) is included in the Nomenclature of Territorial Units for Statistics (NUTS) of the European Union. The two defined NUTS2 regions since (2012 to 2021) was Adriatic and Continental Croatia. Since 2021, Croatia is divided to 4 (non-administrative) regions where Adriatic Croatia remained the same region, and continental Croatia was divided into three ones: Pannonian and Northern Croatia and the city of Zagreb) [7,8]. This is the reason why the characteristics of Croatian population is presented based on two regions (Table 1), where statistically significant differences on the regional level were detected for MD followers (in the largest age group (65–74 years old), the employment status, CMDs (absence of any) and those with the lowest outcome for the maximal of grip strength measure. Relating the regions with food preferences of the inhabitants, it is certainly interesting to study the similarities and/or differences in the two regions; Adriatic and Continental, because the first should definitely have a Mediterranean character, as well as to compare them with other countries in the region.

In the world population the prevalence of overweight and obesity is increasing in all age groups, including the elderly [34,35]. Therefore, the concern for this population group is growing because it is known that obesity is associated with serious medical complications and in the elderly can accelerate age-related decline in physical function, while the underweight is related with some health hazards, such as reduced bone density and fractures [36]. Normal BMI of elderly is associated with significant improvements in quality of life, physical function and health [37], therefore Table 3 and Table 4 and the Figure 2A,B are dedicated to the review of the normally nourished in all countries with separate data for Croatia and the review of the share of the normally nourished and other nutrition groups, according to gender and age groups. The share of obese is higher in the female population, while the overweight is dominant for the male population, and the concerning nourishment group that raises with the age is the underweighted. Studies show association of the lower household income and national income as well, with population obesity prevalence [38,39] and it seems that in Croatia this is one of the factors of the extremely high proportion of overweight and obese people in the observed population group. The share of underweighted in EU countries is under the share indicated for Africa where nearly one among five older people was undernourished [35] and about 30% of African elderly population were overweight or obese [35] while in the EU population, it is over 50% for all age groups. In the Croatian population were not detected any significant differences in average values for different gender, while it is obvious that the average values fall after the age of 85 years, but not significantly (*p* > 0.05).

Finally, in order to confirm or dismiss the positive effect, of following the Mediterranean diet, on health indicators as well as on well-being indicators, logistic regression models were used, which were adjusted considering additional indicators of the observed group such as age, gender, marital status, education level, employment and economic status. Previous investigations [13,17] confirmed the influence of education, retirement and economical status on the outputs as health and well-being, therefore those variables were used in the model adjustment. So, the MD adherence is related to the “et least self-perceived” health as good, CASP at least as Medium, with no regular vigorous sport activities. At least one or more CDMs will make the Mediterranean way of eating more attractive because Mediterranean diet is associated with better cardiovascular health outcomes, including clinically meaningful reductions in rates of coronary heart disease, ischemic stroke, and total cardiovascular disease [40]. The final parameter hand grip strength showed increased significantly relations with the MD adherence what is in accordance with studies investigating the association of adherence to a Mediterranean diet with BMI, muscle strength of adults with diabetes [41], older adult women [42] or in the Asian population [43,44]. Hence, the transition to a healthier diet, such as the Mediterranean diet, is related to the well-being and health of the population, therefore this study points to important factors that should be changed, as a whole, because failure to correct (increase) the mentioned indicators of health and well-being will result additional deterioration of the health of the mature and elderly population. Although the frequency of consumption of the food group fruit and vegetables does not differ significantly for three different European regions, it is important to note the published results of Eurostat in which income plays an important role in the consumption of this group of foods [34]. Consumption of foods from the group’s Dairy products and Fruits and vegetables, of Croatian elderly population (Adriatic and Continental Croatia) is in accordance with the trend of Southern, Western and Central and Eastern Europe countries (Table A4). However, in both Croatian regions, the regular consumption of foods from the group of Legumes, beans, eggs and Meat, fish, poultry is significantly different from the European trend. Research conducted in Croatia, which included socially organized nutrition and the relation with the Mediterranean diet pattern, confirmed that menus in Adriatic region showed the Mediterranean pattern, while Continental ones did not [45]. However, in the characterization of the Mediterranean diet of the elderly population in Croatia (Table 2), an almost negligible share of people from the continental part of Croatia, who follow the Mediterranean diet, was singled out. Although the continental part of Croatia was more represented in the number of participants (919 vs. 297 participants), it should also be noted that precisely in the continental part, especially in the city of Zagreb, there is a significantly larger number of institutions that offer organized nutrition and accommodation for the elderly, and the Mediterranean diet pattern is thus represented in the group of those people who use such accommodation. In two southern European countries, as the traditional diet of northern Portugal and northwestern Spain, the South European Atlantic Diet (SEAD) is present, and adherence to the SEAD has been shown to be associated with a lower risk of death from all causes among the elderly in Spain [46]. Eurostat data [34,47] confirm the findings of regional diversity, which is also reflected in the frequency of certain diseases. Data published in 2019 showed that more that 83% of all deaths in Europe occurred among the elderly (people aged 65 years and over) and the standardized death rate from circulatory diseases was almost five times higher in Bulgaria (Central and Eastern country) than in Spain (Southern country) [43]. Cancer as cause of death is most common in Hungary (327.7 per 100,000 inhabitants) and Croatia (310.9 per 100,000 inhabitants) as a part of the region (Central and Eastern countries) with the highest death rate of 285.5 per 100,000 inhabitants), followed by the Northern countries (259.8 per 100,000 inhabitants), Western countries (237.5 per 100,000 inhabitants) and Southern countries (222.3 per 100,000 inhabitants) [47]. Mediterranean diet pattern rebounds by positive contribution to health status, representing the best nutritional strategy for obtaining a great combination of nutrients, antioxidants and other beneficial molecules able to promote the healthy aging process [48]. Independently of the above, we conducted an additional analysis of the Mediterranean diet and all observed diseases, and we determined the dominance of the Mediterranean diet in those diseases that are most prevalent in the European population as well in the Croatian. Namely participants who were following MD pointed out as their health problem (i) long-term illness (63.7%), (ii) high blood pressure or hypertension (57.7%) and (iii) diabetes (49.6%). Bearing in mind that the Mediterranean diet has positioned itself in a leading position in diet therapy [49,50], the presence of the Mediterranean dietary pattern is clearer in the continental part of Croatia as well, and in general among the population that wants to maintain their health status through diet.

## 5. Conclusions

Our findings confirmed the association of the studied indicators of health and general well-being with MD adherence, for people over 50 years of age. Our study extended the currently available literature covering 27 European countries, placing the findings in a wider geographical setting, however, with the same aspiration to achieve the ultimate goal of caring for the middle-aged and elderly population, their health and general well-being. The Mediterranean diet has once again proven to be an excellent aid in maintaining health. The results of the study also established a positive correlation between adherence to the Mediterranean diet with CASP and self-perceived health, which is predominantly assessed as “very good” or “excellent” by people following the MD regimen. A higher measure of grip strength was found precisely in the MD followers. Data for EU countries, including the data for the Republic of Croatia, have the same trends, but unfortunately, the Republic of Croatia deviates from the average in terms of the proportion of overweight and obese people. All the above results are extremely important for public health services, indicating possible critical factors in maintaining the health of this population.

## Figures and Tables

**Figure 1 ijerph-20-04893-f001:**
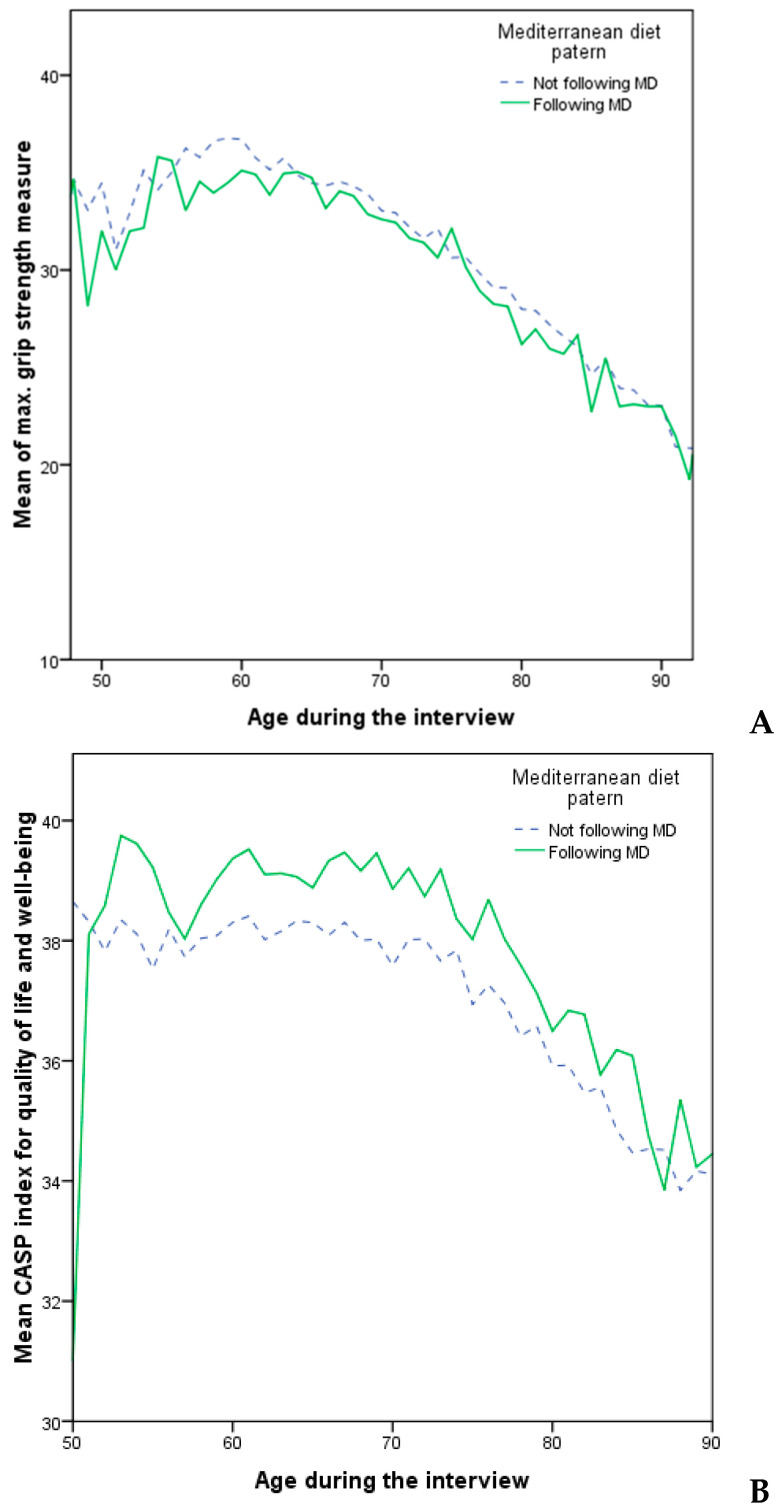
Linkage of (**A**) max. of grip strength measure and (**B**) index for quality of life and well-being (CASP) for those participants (aged 50–90 years) who follow (or not follow) the patterns of the Mediterranean diet.

**Figure 2 ijerph-20-04893-f002:**
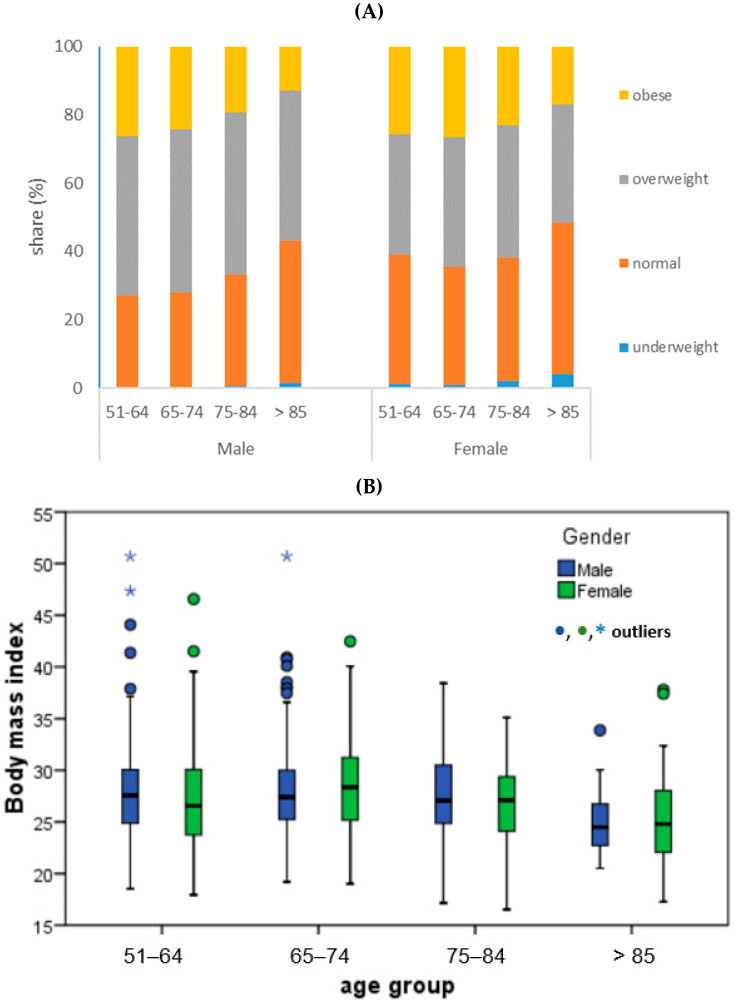
Distribution of the BMI categories based on age and gender, as indication of nourishment of the total participant set (**A**) as well as for Croatia (**B**).

**Table 1 ijerph-20-04893-t001:** Characteristics of the population due to following or not following the Mediterranean diet.

Variables	Not Following MD	Following MD
Age		
50–64	26.26 ^a^	24.35 ^a^
65–74	38.12 ^b^	40.31 ^b^
75–85	26.04 ^a^	26.44 ^a^
>85	9.58 ^c^	9.20 ^c^
Gender		
Female	56.97 *^a^	61.36 *^a^
Male	43.03 ^b^	38.64 ^b^
Marital status		
Living with partner	61.67 *^a^	66.97 *^a^
Not living with partner	38.33*^b^	33.03*^b^
Educational level		
Primary	15.49 ^a^	13.50 ^a^
Secondary	53.00 ^b^	54.16 ^b^
Tertiary	30.44 ^c^	30.69 ^c^
Employment		
Retired	68.88 ^a^	68.49 ^a^
Not retired	31.11 ^b^	31.50 ^b^
Economic status		
Poor	27.52 ^a^	31.58 ^a^
Fair	40.82 *^b^	34.21 *^a^
Good	31.65 ^a^	34.21 ^a^
Self-perceived health		
Poor/Fair	40.32 *^a^	24.55 *^a^
Good	38.03 ^a^	38.35 ^b^
Very good/Excellent	21.55 ^b^	37.05 *^b^
CASP index for quality of life and well-being		
Low	3.26 ^a^	2.11 ^a^
Medium	38.73 ^b^	33.03 ^b^
High	58.01 *^c^	64.89 *^c^
Sport activities that are vigorous		
Never or rare	55.51 ^a^	51.18 ^a^
At least once a week	44.48 ^b^	48.20 ^b^
CMD		
one or more diagnoses	61.61 ^a^	59.99 ^a^
none	38.39 ^b^	40.01 ^b^
Body mass index		
Normal (age 51–64)	32.57	34.13
Normal (age > 65)	49.35	51.93
Max. of grip strength measure		
Low	26.86 ^a^	27.88 ^a^
Medium	66.29 ^b^	66.44 ^b^
High	8.38 ^c^	6.85 ^c^

*—indicates statistically significant differences (*p* < 0.05) in the row for following or not following MD. Different letters in the column indicate statistically significant differences (*p* < 0.05) for different categories of the same variable.

**Table 2 ijerph-20-04893-t002:** Characteristics of the Croatian population (n = 1197) due to following or not following the Mediterranean diet in two NUTS regions (Adriatic (*n* = 297) and Continental (*n* = 910) Croatia).

Variable	Not Following MD		Following MD	
	Adriatic	Continental	Adriatic	Continental
Age				
50–64	31.69 *^a^	34.26 ^a^	44.44 *^a^	31.53 ^a^
65–74	36.62 *^a^	36.02 ^a^	22.22 *^#b^	34.23 ^#a^
75–85	26.06 ^b^	23.22 ^b^	22.22 ^b^	27.93 ^b^
> 85	5.63 ^c^	4.66 ^c^	11.11 ^c^	6.31 ^c^
Gender				
Male	43.66 *^a^	42.94 ^a^	22.22 *^#a^	44.14 ^#a^
Female	56.34 *^b^	57.05 ^b^	77.78 *^#b^	55.86 ^#b^
Marital status				
Living with partner	9.99 ^#a^	43.33 *^#a^	4.91 ^#a^	70.59 *^#a^
Not living with partner	90.01 ^b^	56.67 *^b^	95.09 ^b^	29.41 *^b^
Employment				
Retired	72.53 *^a^	69.37 *^a^	66.67 *^#a^	78.89 *^#a^
Not retired	27.47 ^b^	30.63 *^b^	33.33 ^#b^	21.11 *^#b^
Self-perceived health				
Excellent/very good	13.70 *^a^	13.60 ^a^	24.44 *^a^	18.01 ^a^
Good	32.22 *^b^	36.02 *^b^	24.44 *^a^	27.03 *^b^
Poor/Fair	50.59 ^c^	46.85 ^c^	51.11 ^c^	49.54 ^c^
CASP				
Low	1.45 *^a^	4.16 ^a^	0.0 *^a^	2.75 ^a^
Medium	40.58 ^b^	41.69 ^b^	44.44 ^b^	45.87 ^b^
High	57.97 *^c^	54.16 ^c^	55.56 *^c^	51.38 ^c^
Sport or activities that are vigorous				
Never or rare	70.42 *^#a^	48.74 ^#a^	44.44 *^a^	48.65 ^a^
At least once a week	29.58 *^#b^	51.26 ^#a^	55.56 *^b^	51.35 ^a^
CMD				
0	52.11 ^#a^	30.61 ^#a^	55.56 ^#a^	31.53 ^#a^
1–5	47.89 ^a^	69.39 ^b^	44.44 ^b^	68.47 ^b^
BMI				
Normal (age 51–64)	28.17 *	24.94 *	55.56 *^#^	31.53 *^#^
Normal (age > 65)	45.71 *	43.77 *	66.67 *	51.38 *
Max. of grip strength measure				
Low	22.31 *^a^	29.19 ^a^	37.5 *^#a^	24.76 ^#a^
Medium	65.38 ^b^	60.70 ^b^	62.5 ^b^	61.90 ^b^
High	11.54 ^c^	10.01 ^c^	0.0 ^c^	13.33 ^c^

*–indicates statistically significant differences (*p* < 0.05) in the row for same NUTS region.; #–indicates statistically significant differences (*p* < 0.05) in the row for different NUTS region. Different letters in the column (a, b, c) indicate statistically significant differences (*p* < 0.05) for different categories of the same variable.

**Table 3 ijerph-20-04893-t003:** Frequency and percentage of the Body Mass Index (BMI), regarding the defined categories based on age.

BMI Categories	Age (Years)	BMI Categories *	Age (Years)		
	51–64		65–74	74–85	>85
<18.5–underweight	0.94	<18.5–severe underweight	3.09	5.06	12.32
		18.5–20.9–underweight	4.31	5.01	7.02
18.5–24.9–normal	33.40	21–27.4–normal	48.33	51.50	52.91
25–29.9–overweight	39.67	27.5–30.9–overweight	24.15	22.06	17.78
>30–obese	25.99	31–39.9–obese	18.33	15.44	9.44
		>40–morbid obesity	1.78	0.92	0.50

* BMI categories according ESPEN guidelines.

**Table 4 ijerph-20-04893-t004:** Share of population with the Body mass index in the range defined as normal for a certain age (age 50–64 *: 18.5–24.9 kg/m^2^; age 65+: 21–27.5 kg/m^2^).

Country	N	50–64 *	65–74	75–84	85+
Austria	1567	35.8	51.6	54.6	60.7
Germany	2877	35.4	51.2	57.3	61.8
Sweden	2355	40.4	61.2	62.4	58.7
Netherlands	1905	41.8	57.5	60.9	63.7
Spain	2117	33.2	46.3	47.2	44.2
Italy	2168	38.8	59.7	54.4	52.9
France	2478	42.2	50.5	53.6	48.3
Denmark	2168	38.8	54.5	59.8	55.2
Greece	2996	32.2	54.4	50.4	53.8
Switzerland	1904	48.4	57.3	58.5	57.8
Belgium	2003	40.2	50.1	53.9	54
Israel	927	35.8	53.8	51.3	44.3
Czech Republic	2713	25.3	40.4	43.2	54.3
Poland	2075	26.6	40.7	47.2	44.2
Luxembourg	955	34.8	51.0	45.7	46.7
Hungary	773	27	43.5	51.3	56.5
Slovenia	2496	29	43.2	48.7	58.5
Estonia	3027	25.9	38.6	45.7	52.1
Croatia	1188	30.3	42.4	44.2	55.2
Lithuania	1436	27.2	39.4	46.3	57.3
Bulgaria	907	30.7	39	54.4	45.2
Cyprus	537	35.2	40	45	44.8
Finland	1161	30.5	51.7	55.5	71.4
Latvia	795	24.3	38.8	44	42
Malta	805	14.6	23.8	30.2	24.4
Romania	1280	24.6	39.7	39.3	38.6
Slovakia	997	30.5	48.2	45.6	57.1

* standard BMI categories; for age 65+: according to ESPEN guidelines.

**Table 5 ijerph-20-04893-t005:** Adjusted odds ratios (95% CI) for Mediterranean diet adherence.

Variables	Following MD
Self-perceived health	
Poor/Fair	0.921 [0.846; 1.003]
Good	0.927 [0.859; 1.000] *
Very good/Excellent	1.541 [1.586; 1.499]
CASP index for quality of life and well-being	
Low	1.095 [0.406; 2.952] **
Medium	1.158 [0.756; 1.776] **
High	1.195 [0.406; 2.952]
Sport activities that are vigorous	
Never or rare	0.942 [0.28; 3.17]
At least once a week	0.91 [0.617; 1.341] **
CMD	
one or more diagnoses	1.019 [0.957; 1.085]
none	0.981 [0.922; 1.045]
Body mass index	
Normal	1.054 [0.993; 1.119]
Max. of grip strength measure	
Low	0.774 [0.689; 0.869] **
Medium	1.449 [1.28; 1.641] **
High	1.293 [1.151; 1.452] *

Adjusted for age, gender, marital status, education level, employment and economic status. *: *p* < 0.05; **: *p* < 0.01

## Data Availability

Not applicable.

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
