# Peer review of "Adherence to the Mediterranean Diet Related to the Health Related and Well-Being Outcomes of European Mature Adults and Elderly, with an Additional Reference to Croatia"

_ijerph, 2023, doi:10.3390/ijerph20064893_

Round 1

Reviewer 1 Report

The authors took up the current problem of maintaining the aging population of Europe in good health. They pointed out using the Mediterranean diet as one of the ways to improve the health of the elderly. It was possible to confirm the significant impact of the Mediterranean diet on the increase in self-assessment of health and grip strength and maintaining proper BMI. It is a pity that the authors devoted little space to an essential part of the study. An interesting issue could be examining the relationship between the use of the Mediterranean diet and, for example, life expectancy or the frequency of various lifestyle diseases in Western, Northern, and Central European countries. These would be arguments confirming the desirability and effectiveness of the Mediterranean diet.

Author Response

Reviewer 1

First of all, thanks for all the comments and suggestions and the detailed clarification of the ambiguities. I thank you on behalf of myself and all the authors.

We agree with all suggested, and we have made changes in the paper itself, and here follow the answers with the indication of the lines where the changes can be seen.

The authors took up the current problem of maintaining the aging population of Europe in good health. They pointed out using the Mediterranean diet as one of the ways to improve the health of the elderly. It was possible to confirm the significant impact of the Mediterranean diet on the increase in self-assessment of health and grip strength and maintaining proper BMI. It is a pity that the authors devoted little space to an essential part of the study. An interesting issue could be examining the relationship between the use of the Mediterranean diet and, for example, life expectancy or the frequency of various lifestyle diseases in Western, Northern, and Central European countries. These would be arguments confirming the desirability and effectiveness of the Mediterranean diet.

A: Yes, we agree that the association of MD with life style in EU regions (Northern; Western; Southern & Central and Eastern) and disease incidence  would be further confirmation of the importance of such a dietary pattern.

Therefore, we used the EuroVoc  classification and divided countries form the SHARE database calculating the consumption , and additionally included data from the SHARE database related to the most common diseases. Those results are presents in supplementary tables S4 and S5. Also added text in "Results" (L397-417) and "Discussion" (L489-532).

We believe that this correction greatly contributed to what was the basic idea of the manuscript-relation between the nutritional pattern and healthiness (and diseases) in the observed population group

Reviewer 2 Report

The paper provides a fairly detailed overview of the relationship between adherence to the Mediterranean diet (MD) and variations in individuals' health outcomes (e.g. incidence of chronic diseases), their body mass index, and level of self-assessed well-being in the Republic of Croatia. The Mediterranean diet is known to be associated with better cardiovascular health outcomes, including clinically significant reductions in rates of coronary heart disease, ischemic stroke, and overall cardiovascular disease, and the results presented in the study confirm the link between adherence to MD and health and well-being outcomes. The data of the project SHARE (more than 114,000 participants), in which 27 European countries and Israel participated, were used in the study. The investigation covered the Adriatic and continental regions of Croatia. According to this study, the characteristics of the population in terms of adherence to MD show statistically significant differences in terms of gender (preferentially women), living with a partner, adequate economic status, and high CASP index, as well as for participants who self-rated their health as very good/excellent. Similar results were confirmed, with the difference that CMDs were negatively associated with adherents of MD. Additionally, the share of obesity is higher in the female population, while overweight is dominant for the male population, and the concerning nourishment group that rises with the age is underweighted.

The paper is generally well-written and structured, but in my opinion, there are some interesting findings that could be discussed further:

What is surprising about the results obtained is that there are no such big differences between the Adriatic and Continental regions in terms of the percentage of participants who do or do not adhere to MD. It was perhaps to be expected that the percentage of respondents in the Adriatic region following MD would be higher than in the continental region, as fish and seafood are traditionally much more common in the diet of people in the Adriatic region. In contrast, the results of the study showed that a higher percentage of the population in Continental Croatia (age groups 65-74 and 75-85) followed MD compared to the Adriatic region. What could be the reason for this? Is it because of the greater purchasing power in the continental region? Is it because the continental region includes the capital city Zagreb with an urban population that has a greater awareness of the importance of a healthy diet? Please try to explain.

In the Adriatic region, 77.78% of women and only 22.22% of men follow MD, while in the continental region, the ratio is different (55.86% of women and 44.14% of men). What could be the reason for these differences? Furthermore, among participants from the two regions following MD, there is a drastic difference in the percentage of those living with a partner (4.91% in the Adriatic region and 70.59% in the Continental region). What could be the reason for this? To what extent can marry life influence eating habits? Was the sample representative, i.e. were there the same number of respondents living with a partner in both regions?

The authors concluded that the Republic of Croatia deviates from the average in terms of the percentage of overweight and obese people obtained from other EU countries. What could be the reason for this? Is it a question of the poorer standard of living in Croatia, which causes a poorer quality of nutrition? Or is it a question of the culture of the people in Croatia, which is such that they do not pay much attention to healthy living and nutrition? Or is it due to an insufficiently developed awareness of the need for a healthy diet? Please try to discuss this.

Finally, the results obtained are very important for public health services, as they highlight possible critical factors for maintaining the health of the population over 50 years of age. Therefore, after a thorough revision, I propose the paper for publication.

Author Response

Reviewer 2

First of all, thanks for all the comments and suggestions and the detailed clarification of the ambiguities. I thank you on behalf of myself and all the authors.

We agree with all suggested, and we have made changes in the paper itself, and here follow the answers with the indication of the lines where the changes can be seen.

The paper provides a fairly detailed overview of the relationship between adherence to the Mediterranean diet (MD) and variations in individuals' health outcomes (e.g. incidence of chronic diseases), their body mass index, and level of self-assessed well-being in the Republic of Croatia. The Mediterranean diet is known to be associated with better cardiovascular health outcomes, including clinically significant reductions in rates of coronary heart disease, ischemic stroke, and overall cardiovascular disease, and the results presented in the study confirm the link between adherence to MD and health and well-being outcomes. The data of the project SHARE (more than 114,000 participants), in which 27 European countries and Israel participated, were used in the study. The investigation covered the Adriatic and continental regions of Croatia. According to this study, the characteristics of the population in terms of adherence to MD show statistically significant differences in terms of gender (preferentially women), living with a partner, adequate economic status, and high CASP index, as well as for participants who self-rated their health as very good/excellent. Similar results were confirmed, with the difference that CMDs were negatively associated with adherents of MD. Additionally, the share of obesity is higher in the female population, while overweight is dominant for the male population, and the concerning nourishment group that rises with the age is underweighted.

The paper is generally well-written and structured, but in my opinion, there are some interesting findings that could be discussed further:

What is surprising about the results obtained is that there are no such big differences between the Adriatic and Continental regions in terms of the percentage of participants who do or do not adhere to MD. It was perhaps to be expected that the percentage of respondents in the Adriatic region following MD would be higher than in the continental region, as fish and seafood are traditionally much more common in the diet of people in the Adriatic region. In contrast, the results of the study showed that a higher percentage of the population in Continental Croatia (age groups 65-74 and 75-85) followed MD compared to the Adriatic region. What could be the reason for this? Is it because of the greater purchasing power in the continental region? Is it because the continental region includes the capital city Zagreb with an urban population that has a greater awareness of the importance of a healthy diet? Please try to explain.

A: Thanks for this extremely important suggestion. Yes, the above is quite confusing, and we have added further clarification in the text. Please see additional tables S4 and S5 as well as the added text (L397-417 & L489-532).

In the Adriatic region, 77.78% of women and only 22.22% of men follow MD, while in the continental region, the ratio is different (55.86% of women and 44.14% of men). What could be the reason for these differences? Furthermore, among participants from the two regions following MD, there is a drastic difference in the percentage of those living with a partner (4.91% in the Adriatic region and 70.59% in the Continental region). What could be the reason for this? To what extent can marry life influence eating habits? Was the sample representative, i.e. were there the same number of respondents living with a partner in both regions?

A: Thank you for this comment because this topic is an important issue.

Firstly, the number of the participants is 3 times larger in the Continental part (this was commented in the L504). In order to justify the disproportionate share of men and women, text was added that the majority of participants who stated that they did not live with a partner were widowed, and the life expectancy of women is also longer (please see L329-331).

The authors concluded that the Republic of Croatia deviates from the average in terms of the percentage of overweight and obese people obtained from other EU countries. What could be the reason for this? Is it a question of the poorer standard of living in Croatia, which causes a poorer quality of nutrition? Or is it a question of the culture of the people in Croatia, which is such that they do not pay much attention to healthy living and nutrition? Or is it due to an insufficiently developed awareness of the need for a healthy diet? Please try to discuss this.

A: Thank you for this comment, we have  because this topic is an important issue. We have added two studies that confirm the relationship of the household income and national income with the prevalence of obesity, in order to explain the high proportion of obese people in the Republic of Croatia. Please see text in L460-463.

Finally, the results obtained are very important for public health services, as they highlight possible critical factors for maintaining the health of the population over 50 years of age. Therefore, after a thorough revision, I propose the paper for publication.

A: Thank you for all the suggestions that enabled additional clarification in monitoring the eating habits and health of this population group

Reviewer 3 Report

Adherence to the Mediterranean diet related to the health related and well-being outcomes of European mature adults and elderly, with an additional reference to Croatia

Comments:

The abstract needs to be completely reorganized: typically, it should begin with a statement or two introductory to the topic. Then mention the lack of existing research or need to carry out the study, a description of the study’s objective, and the methods carried out to achieve it. Describe the most relevant findings of the study and future prospects.

Introduction

Lines 46-47. Please improve the wording. An increase in disease (in the general population and the elderly) may be due to environmental changes (toxic substances, pandemics, etc.) and not necessarily be related to people's healthy behavior.

Line 67 “Obesity caused by a mutation of the melanocortin receptor increases mortality in various ways” If what the authors want to highlight is that obesity increases risk factors for mortality, this statement should be modified, since obesity has a complex etiology (not only genetic), and for the purposes of this study, the statement is out of place.

Line 72. “But in certain age increases the share of malnourished population”. Change the word “But” to “It is also known that” since the information that precedes this statement does not conflict with current knowledge. Overweight or obesity can coexist with vitamin deficiencies, for example.

Line 82 “Importance of consumption of certain food groups is confirmed to have a strong effect on quality of life” Please develop this statement further, as it remains ambiguous. An example or two of this association would suffice.

Line 85. Take advantage of the Introduction section to explain what the Mediterranean diet is and how it is characterized. The Mediterranean diet is the central theme of this study and must be explained before being mentioned.

Lines 84-87. Please add the geographical delimitation of the study so that it makes more sense in your manuscript.

Based on this direct history, the need for the study is not entirely clear. It is recommended that before stating the research objective, a statement of the problem be added, along with information the lack of existing research which justifies the need for this study.

Material and Methods

Main methodological limitations: The Mediterranean diet is not precisely characterized, and therefore, what is being analyzed are partial "adherence" data. Key foods in the Mediterranean diet, such as seeds and oilseeds, the consumption of olive oil, red wine, and seafood (separate from red meat and sausages) are not taken into account. Likewise, a binary scale of compliance or non-compliance with the Mediterranean diet was provided without attempting to establish categories. This is a significant limitation of the study.

Author Response

Reviewer 3

First of all, thanks for all the comments and suggestions and the detailed clarification of the ambiguities. I thank you on behalf of myself and all the authors.

We agree with all suggested, and we have made changes in the paper itself, and here follow the answers with the indication of the lines where the changes can be seen.

Comments:

The abstract needs to be completely reorganized: typically, it should begin with a statement or two introductory to the topic. Then mention the lack of existing research or need to carry out the study, a description of the study’s objective, and the methods carried out to achieve it. Describe the most relevant findings of the study and future prospects.

A: The summary has been modified according to your suggestions. Please see L17-60.

Introduction

Lines 46-47. Please improve the wording. An increase in disease (in the general population and the elderly) may be due to environmental changes (toxic substances, pandemics, etc.) and not necessarily be related to people's healthy behavior.

A: The sentence has been modified to emphasize that an individual's behavior is only one of the factors that affect his general well-being. Please see L70-76.

Line 67 “Obesity caused by a mutation of the melanocortin receptor increases mortality in various ways” If what the authors want to highlight is that obesity increases risk factors for mortality, this statement should be modified, since obesity has a complex etiology (not only genetic), and for the purposes of this study, the statement is out of place.

A: We agree. This sentence is deleted. (L96-97)

Line 72. “But in certain age increases the share of malnourished population”. Change the word “But” to “It is also known that” since the information that precedes this statement does not conflict with current knowledge. Overweight or obesity can coexist with vitamin deficiencies, for example.

A: We agree. The suggested phrase was used to replace “But” (please see L102)

Line 82 “Importance of consumption of certain food groups is confirmed to have a strong effect on quality of life” Please develop this statement further, as it remains ambiguous. An example or two of this association would suffice.

A: We agree that this statement needs to be clarified. Therefore, we added new text, please see L112-122.

Line 85. Take advantage of the Introduction section to explain what the Mediterranean diet is and how it is characterized. The Mediterranean diet is the central theme of this study and must be explained before being mentioned.

A:We agree. This is included in the corrections in L112-122.

Lines 84-87. Please add the geographical delimitation of the study so that it makes more sense in your manuscript.

A: Thank you for the comment. Now it is added (please see L124).

Based on this direct history, the need for the study is not entirely clear. It is recommended that before stating the research objective, a statement of the problem be added, along with information the lack of existing research which justifies the need for this study.

A: The aim of the study is more clearly explained. Please see L128-133.

Material and Methods

Main methodological limitations: The Mediterranean diet is not precisely characterized, and therefore, what is being analyzed are partial "adherence" data. Key foods in the Mediterranean diet, such as seeds and oilseeds, the consumption of olive oil, red wine, and seafood (separate from red meat and sausages) are not taken into account. Likewise, a binary scale of compliance or non-compliance with the Mediterranean diet was provided without attempting to establish categories. This is a significant limitation of the study.

A: We fully agree, and we placed that note (limitation of this approach) at the end of subchapter 2.2.

The binary scale is a limitation, but we have now added the frequencies of food consumption in four different European regions, to minimize this limitation (please see added table S4).

Round 2

Reviewer 2 Report

I find that the authors have answered the questions adequately and corrected the paper according to the suggestions. Therefore, ​I propose the manuscript for publication in present form.

Reviewer 3 Report

Most of the observations/comments I have made to the original manuscript have been answered to my satisfaction.